:Ö: PLOS | ONE

# Centromeres of *Cucumis melo* L. comprise *Cmcent* and two novel repeats, *CmSat162* and *CmSat189*

**Agus Budi Setiawan**[1¤], **Chee How Teo**[2], **Shinji Kikuchi**[1], **Hidenori Sassa**[1], **Kenji Kato**[3], **Takato Koba**[1]*

**1** Laboratory of Genetics and Plant Breeding, Graduate School of Horticulture, Chiba University, Matsudo, Chiba, Japan, **2** Center for Research in Biotechnology for Agriculture, University of Malaya, Kuala Lumpur, Malaysia, **3** Graduate School of Environmental and Life Science, Okayama University, Kita-ku, Okayama, Japan

¤ Current address: Laboratory of Genetics and Plant Breeding, Faculty of Agriculture, Universitas Gadjah Mada, Jalan Flora Bulaksumur, Yogyakarta, Indonesia
* koba@faculty.chiba-u.jp

**Data Availability Statement:** All relevant data are within the manuscript and its Supporting Information files.

## Abstract

Centromeres are prerequisite for accurate segregation and are landmarks of primary constrictions of metaphase chromosomes in eukaryotes. In melon, high-copy-number satellite DNAs (SatDNAs) were found at various chromosomal locations such as centromeric, pericentromeric, and subtelomeric regions. In the present study, utilizing the published draft genome sequence of melon, two new SatDNAs (*CmSat162* and *CmSat189*) of melon were identified and their chromosomal distributions were confirmed using fluorescence *in situ* hybridization. DNA probes prepared from these SatDNAs were successfully hybridized to melon somatic and meiotic chromosomes. *CmSat162* was located on 12 pairs of melon chromosomes and co-localized with the centromeric repeat, *Cmcent*, at the centromeric regions. In contrast, *CmSat189* was found to be located not only on centromeric regions but also on specific regions of the chromosomes, allowing the characterization of individual chromosomes of melon. It was also shown that these SatDNAs were transcribed in melon. These results suggest that *CmSat162* and *CmSat189* might have some functions at the centromeric regions.

## Introduction

Repetitive DNA sequences form a major portion of nuclear DNA in eukaryotic genomes, particularly in melon, accounting for 42% of the total sequence [1,2]. Repetitive DNA sequences are organism-specific at the species or genus level and/or chromosomal location-specific including centromeric or subtelomeric [3]. Repetitive DNAs are classified into two major groups, namely, tandem repeats (micro-, mini-, or satellite DNA) and dispersed repeats, i.e., transposable elements (DNA transposons and retrotransposons). Tandem repeats are arranged in tandem arrays of monomeric units [4], whereas dispersed repeats are mobile and

**Funding:** This work was financially supported in part by Japan Society for Promotion of Science (JSPS) KAKENHI Grant Number JP 16K07588 to TK. The funders had no role in study design, data collection and analysis, decision to publish, or preparation of the manuscript.

**Competing interests:** The authors have declared that no competing interests exist.

are scattered across the genome [5,6]. Satellite DNA (SatDNA) families are in a special class of tandemly repeated monomers in heterochromatic regions comprising 150–400 base pairs (bp) of DNA [3,4,7].

Melon (*Cucumis melo* L.) belongs to Cucurbitaceae family and is a diploid species possessing $2n = 2x = 24$ chromosomes [8]. The relatively large amount (20%-30%) of SatDNAs in Cucurbitaceae serves as an interesting resource for the identification of new SatDNA [9]. pSat107 is a melon-specific SatDNA with a nucleotide sequence length of 352 bp [10] and it hybridizes to melon centromeres [11,12,13]. Centromeres are important for sister chromatid segregation during cell division. Heterochromatic regions are characterized as those with accumulation of SatDNAs and favorable sites for centromeres [14]. Plant centromeres are composed of satellite DNA repeats and highly repeated centromere-specific retrotransposons [15]. Functional centromeres are determined by the occurrence of nucleosomes containing centromere-specific histone H3 (CENH3), the binding of which to DNA can be analyzed by chromatin immunoprecipitation [16,17,18,19,20].

The melon genome is 454 mega-base pairs (Mb) in size [1]. To date, only *Cmcent* has been reported as a centromere marker in melon [12], and there are no reports on other centromeric repeats in melon. Utilizing the draft melon genome sequence, we identified two new SatDNAs, namely, *CmSat162* and *CmSat189*, as melon centromeric repeats. Here we describe successful hybridization of them to melon somatic chromosomes and pachytene chromosomes, and their distribution on the chromosomes which can be used for chromosome identification. We also discuss transcription of these SatDNA repeats to identify their possible functions in centromere structures.

## Materials and methods

### Plant materials

Three Indonesian melon cultivars,'Baladewa' (*Cucumis melo* L. subsp. *melo* var. *cantalupo* Ser.), 'Ivory F$_1$ hybrids' (*C. melo* L. subsp. *melo* var. *inodorus*) and 'P90' (*C. melo* L. subsp. *agrestis* var. *conomon*), were used in this study. The seeds were germinated on moistened filter paper in petri dishes and grown in a growth chamber at 25˚C.

### Data mining and dot plot identification of SatDNA repeats

Scaffold sequences of *Cucumis melo* DHL92 (BioProject accession PRJEB68; [21]) were retrieved from the National Center for Biotechnology Information (NCBI) database and subjected to tandem repeat sequence analysis using Tandem Repeat Finder version 4.09 (http://tandem.bu.edu/trf/trf.basic.submit.html). New SatDNA sequences were identified in the melon genomic scaffold sequence "LN681816" (S1 Table). Five tandem repeats were detected, and two of them (*CmSat162* and *CmSat189*) were used. The consensus sequences of these SatDNAs and their sequence alignments are shown in S2 Table, S1 and S2 Figs. The selected sequences were blasted against the melon genome database (http://cucurbitgenomics.org/blast), and the primers were designed using FastPCR software [22].

### Genomic DNA isolation

Genomic DNA was extracted using a modified version of the method of Doyle and Doyle [23]. In brief, 0.5 g of young leaves were ground in 500 μL of cetyltrimethylammonium bromide (CTAB) isolation buffer [2% (w/v) CTAB, 1.4 M NaCl, 20 mM EDTA, 100 mM Tris-HCl pH. 8.0, 0.2% (v/v) 2-mercaptoethanol] that had previously been incubated at 60˚C in a preheated mortar before transferring into 1.5-ml tubes. The samples were incubated at 60˚C for 30 min

in a water bath with occasional gentle swirling. Next, 200 μL of 24:1 (v/v) chloroform:isoamyl alcohol (CIAA) was added and mixed gently but thoroughly, and the mixture was centrifuged at 1600 x g for 15 min at RT followed by removal of the aqueous phase at the top of the tube. Finally, the samples were transferred into a new 1.5-ml tube, and the genomic DNA was purified using CIAA twice. Then, 1/10 volume of 3 M sodium acetate and 2/3 volume of cold isopropanol were consecutively added. The samples were kept at -30˚C for 1 h to precipitate the DNA and centrifuged at 500 x g for 2 min. The supernatant was poured off, followed by the addition of 600 μL of DNA wash buffer (60 mM potassium acetate, 10 mM Tris-HCl pH 7.5, 60% ethanol). Subsequently, the tubes were centrifuged at 10,000 x g for 1 min, and then the supernatant was poured off again. The samples were washed twice with DNA wash buffer and air-dried using the Automatic Environmental SpeedVac System AES1010 for 30 min. The pellet was re-suspended in 100 μL of TE buffer (10 mM Tris-HCl pH 7.4, 1 mM EDTA). One μL of RNase A (Qiagen) at 1 mg/μL was added and incubated at 37˚C for 1 h. Finally, the DNA quality was analyzed by electrophoresis.

## Total RNA isolation

Total RNA was extracted from leaves by a slightly modified version of a method described elsewhere (http://www.patentsencyclopedia.com/app/20090111114). In brief, 0.5 g of each sample was ground into powder in liquid nitrogen in a chilled mortar and pestle with 300 μL of RNA isolation buffer [48% (w/v) guanidine thiocyanate (GTC), 10 mM 2-morpholinoethanesulfonic acid (MES) pH 6.46, 1% (w/v) polyvinylpyrrolidone (PVP), and 0.2% (v/v) 2-mercaptoethanol]. The mixture was transferred into 2-ml tubes to which 500 μL of CIAA was added and mixed gently. The samples were centrifuged at 10,000 x g for 15 min. The aqueous phase was transferred into a PD column, centrifuged at 10,000 x g for 1 min, and the supernatant was discarded. Six hundred μL of 40% diethylene glycol dimethyl ether (Diglyme) was added to the tube before centrifugation at 10,000 x g for 1 min. The supernatant was carefully discarded from the tube to avoid removing the loosely attached pellet. Next, 600 μL of 80% ethanol was added to the tube to purify the pellet, followed by centrifugation at 10,000 x g for 1 min. The washing steps with Diglyme and ethanol were repeated twice. The PD column was centrifuged at 10,000 x g for 1 min to completely dry the column. Finally, 30 μL of pre-warmed sterile distilled water was added and centrifuged at 10,000 x g for 1 min. The total RNA was treated with Deoxyribonuclease RT Grade (Nippon Gene, Japan) to remove the genomic DNA in accordance with the manufacturer's instructions.

## Cloning of satellite DNA repeats

All SatDNA repeats used are listed in Table 1. The monomer length of *CmSat162* and *CmSat189* (162 bp and 189 bp, respectively) were determined based on bioinformatic analysis. These repeats were isolated by polymerase chain reaction (PCR) amplification of 'P90'

**Table 1. SatDNA repeats used in this study.**

| Name | Type | Length (bp) | Source |
|---|---|---|---|
| *CmSat162* | Satellite DNA | 162* | This study |
| *CmSat189* | Satellite DNA | 189* | This study |
| *Cmcent* | Satellite DNA | 354 | Koo et al. 2010 |

*: The monomer length of these SatDNAs were identified using Tandem Repeat Finder from melon scaffold sequence (acc. No. LN681816) deposited in NCBI GenBank.

genomic DNA using the oligonucleotide primer pairs `5'-GGATTGTCGTACTTGAACACTTG GT-3'` and `5'-CCTAAGTAGTGTTCATGAGGTGCCT-3'`, and `5'- CACATCATAACAAGT GTATCAACA-3'` and `5'-TCATCCACGAAGCATGATAC-3'`, respectively. The resulting 311-bp and 267-bp PCR products amplified from the dimers were cloned into pGEM-T-Easy Vector (Promega) in accordance with the manufacturer's protocol.

## Semi-quantitative PCR amplification of *CmSat162* and *CmSat189* transcripts

First-strand cDNAs were synthesized from 0.5 μg of total RNA using ReverTraAce® qPCR RT Master Mix with gDNA Remover (Toyobo, Japan). The resulting cDNA was used as a template in a 30-μl PCR reaction volume using gene-specific primers of *CmSat162* and *CmSat189*. Semi-quantitative PCR (sqPCR) was performed with a PCR Thermal Cycler Dice™ Touch (Takara, Japan) using TaKaRa *Ex Taq* Hot Start Version (TaKaRa, Japan). The sqPCR products were separated on 2% agarose gel and stained with ethidium bromide before visualization using High Performance UV Transluminator (USA). The β-actin gene was used as an internal control for determining the sqPCR amplification efficiency in the tissue samples, and it was amplified using the primer pair PbActin2f1 and PbActin2r1 [24].

## Chromosome and probe preparations and fluorescence *in situ* hybridization (FISH)

The preparations of mitotic metaphase and meiotic pachytene chromosomes were conducted using modified Carnoy's solution II in accordance with the work of Setiawan et al. [13]. The *Cmcent* probe was labeled with Biotin-Nick translation mix (Roche), whereas *CmSat162* and *CmSat189* were labeled with Dig-Nick translation mix (Roche). The FISH protocol as described by Setiawan et al. [13] was followed. For the pachytene chromosomes, the hybridization mixtures were added on the chromosome preparations, covered with a 22 x 40-mm cover slip and sealed with rubber cement. The slides were denatured on a hot plate at 80˚C for 2–3 min. Finally, the slides were placed in a humidity chamber and incubated at 37˚C overnight. Detection solutions of 126 μL [1% BSA in 4x SSC 125 μl + 0.4 μl/ml anti-digoxigenin rhodamine (Roche) 0.5 μL + 0.5 μg/mL biotinylated streptavidin-FITC (Vector Laboratories) 0.5 μl] were used and washed in 2x and 0.1x SSC for 3 min after incubation at 37˚C for 30 min. Finally, the slides were counterstained with 4,6-diamidino-2-phenylindole (DAPI) in a Vecta-Shield antifade solution (Vector Laboratories).

## Sequence comparison and image analysis

The comparison among *Cmcent*, *CmSat162*, and *CmSat189* sequences was performed using a dot plot and analyzed using Unipro UGENE software. Karyotyping ideograms were constructed using CHIAS IV [25]. FISH signals were observed under a fluorescence microscope (Olympus BX53) equipped with a cooled CCD camera (Photometrics CoolSNAP MYO), processed using Metamorph, Metavue imaging series version 7.8, and edited using Adobe Photoshop CS 6.

## Results

### *CmSat162*, a major component of melon centromeres

We conducted physical localization of *CmSat162* and *Cmcent* on somatic metaphase chromosomes, and meiotic pachytene chromosomes. *CmSat162* produced major FISH signals on all melon centromeres, and it co-localized with *Cmcent*, a previously known melon centromere

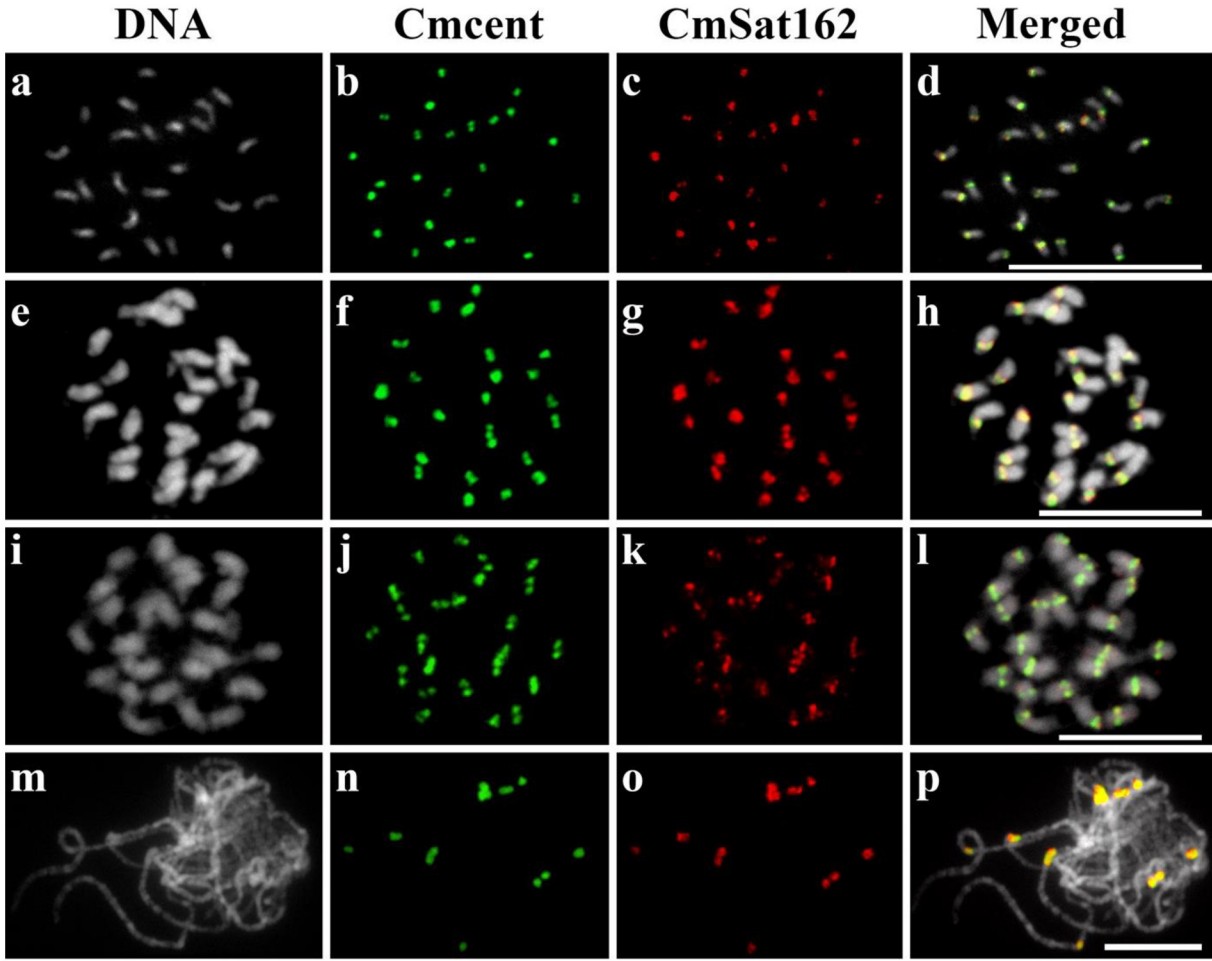

**Fig 1. Physical mapping of SatDNAs on melon mitotic and meiotic chromosomes, and extended DNA fibers using FISH.** Metaphase chromosomes of 'Baladewa' (a-d)), 'Ivory F$_1$ hybrids' (e-h), 'P90' (i-l), and pachytene chromosome of 'P90' (m-p). *Cmcent* (green) and *CmSat162* (red). Scale bars = 10 µm.

marker (Fig 1). *pSat107* and *Cmcent* shared high sequence homology (Fig 3d), and pSat107 was identified to be *Cmcent* [20]. *CmSat162* and *Cmcent* were successfully hybridized on metaphase cells of 'Baladewa', 'Ivory F$_1$ hybrids', and 'P90' (Fig 1d, 1h and 1l). High-resolution FISH on 'P90' pachytene chromosomes was carried out to reveal the organization of these SatDNAs (Fig 1m–1p). Both repeats were located within a heterochromatic block of primary constrictions (Fig 1p). These results suggest that *CmSat162* is an additional new centromeric marker in melon beside *Cmcent*.

### *CmSat189*, a part of the melon centromeres

*CmSat189* was successfully hybridized to all chromosomes of 'Ivory F$_1$ hybrids' (Fig 2). This probe produced major signals on primary constrictions and additional ones on chromosome-specific regions, permitting characterization of individual homologous chromosomes (Fig 2c and 2d). The majority of the signals co-localized with those of *Cmcent* and *CmSat162*, although they shared low sequence homologies (Figs 2a, 2b and 3a–3c, and S3 Fig). Some of them were located at interstitial, pericentromeric, or subtelomeric regions depending on the chromosome pairs (Fig 2d, Table 2); *e.g.*, chromosome 1 was distinguished from others with the presence of

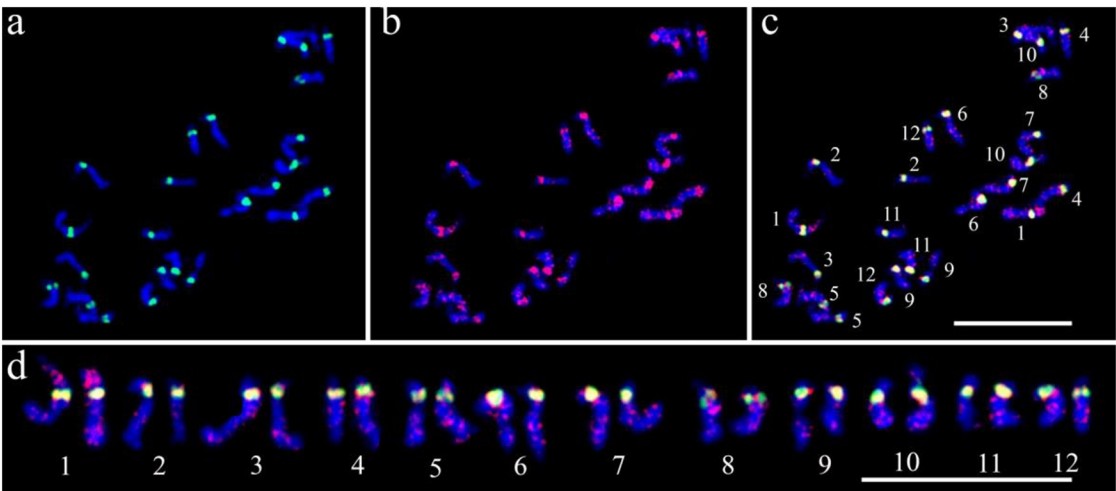

**Fig 2. Physical mapping of *CmSat189* on melon mitotic chromosomes of 'Ivory F₁ hybrids'.** FISH detection of *Cmcent* (a) and *CmSat189* (b) in prometaphase chromosomes. Overlay image of *Cmcent* and *CmSat189* (c). Homologous chromosomes are arranged from left to right in the order of their lengths (No. 1 to No. 12), and based on the locations of *CmSat189* and *Cmcent* signals (d). *Cmcent* (green) and *CmSat189* (red). Scale bars = 10 μm.

the signals of *CmSat189* at the centromeric and interstitial regions of the short arms with strong intensities, at the interstitial region of the long arms with medium intensities, and at the pericentromeric and subtelomeric regions of the long arms with weak intensities (Fig 2d). An ideogram was constructed using prometaphase chromosomes based on the condensation patterns of heterochromatic and euchromatic regions, as well as *Cmcent* and *CmSat189* hybridization signals (Fig 3e).

## Transcription of *CmSat162* and *CmSat189* in melon

The expression of *CmSat162* and *CmSat189* was examined using semi-quantitative PCR with repeat-specific primers (Fig 3f). The result showed that both repeats were expressed in leaf tissue of melon, and the expression level was similar to that of the endogenous control, the β-actin gene. This suggests that both SatDNAs are actively transcribed in melon with some specific unknown functions.

## Discussion

Repetitive sequences, particularly SatDNAs, are found with high copy numbers within eukaryotic genomes. They are primarily located at centromeric, pericentromeric, and telomeric regions, which form the major components of heterochromatin [26]. They play important roles in biological processes related to cellular and chromosomal functions [3,27]. Therefore, understanding the organization and function of SatDNAs should contribute toward accelerating molecular cytogenetic research in higher plants. Furthermore, SatDNAs can be used to identify homologous chromosomes and determine the positions of centromeres in plants [11,12,14,20,28]. Utilizing the melon genome sequence, we identified new SatDNAs and detected their distributions in melon chromosomes.

Centromeres play essential roles in sister chromatid cohesion, and they are predominantly composed of SatDNAs and retrotransposons [15]. In the present study, five tandem repeats were detected from the published DNA sequence LN681816, and two tandem repeats (*CmSat162* and *CmSat189*) were used (S1 Table). The monomer length of *CmSat162* (162 bp) is

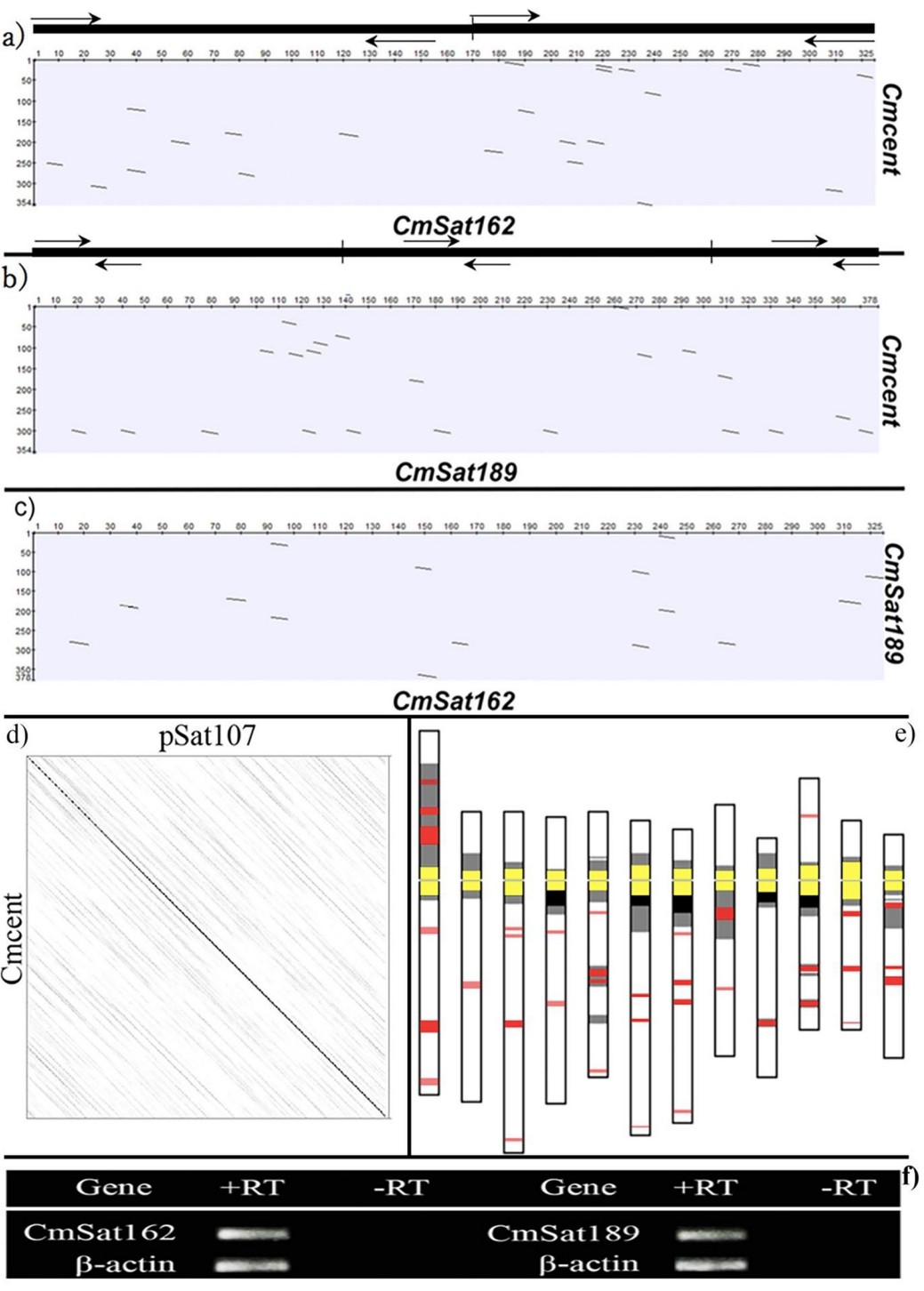

**Fig 3.** Sequence comparison analysis between *Cmcent* and *CmSat162* (a), *Cmcent* and *CmSat189* (b), *CmSat162* and *CmSat189* (c), and Cmcent and pSat107 (d) using dot plot, and expression analysis of *CmSat162* and *CmSat189* amplified from leaf tissues of melon (f). The positions of monomers and PCR primers of *CmSat162* and *CmSat189* are indicated with black horizontal bars and arrows above the dot plot, respectively. Monomer of *Cmcent* was used for the dot plot analysis. Ideogram of melon using prometaphase chromosomes based on condensation patterns, and locations of *Cmcent* and *CmSat189* hybridization signals (e). Centromere signals (yellow) resulted from co-localization of signals of *Cmcent* and *CmSat189*, *CmSat189* signals (red), heterochromatic region (black), and euchromatic region (gray).

**Table 2. Features of melon chromosomes based on *CmSat189* hybridization signals.**

| Chromosome number | Chromosome features based on *CmSat189* hybridization signals |
|---|---|
| 1 | Highest signal intensity at centromeric region, high at interstitial region of 1S, medium at interstitial region of 1L, and weak at pericentromeric and subtelomeric regions of 1L. |
| 2 | Highest signal intensity at centromeric region and weak at interstitial region of 2L. |
| 3 | Highest signal intensity at centromeric region, medium at interstitial region, and weak at pericentromeric and subtelomeric regions of 3L. |
| 4 | Highest signal intensity at centromeric region and weak at pericentromeric region and interstitial regions of 4L. |
| 5 | High signal intensity at centromeric region, medium at interstitial regions, and weak at pericentromeric and subtelomeric regions of 5L. |
| 6 | Highest signal intensity at centromeric region, medium at interstitial region, and weak at pericentromeric region of 6L. |
| 7 | High signal intensity at centromeric region, medium at interstitial region, and weak at pericentromeric and subtelomeric regions of 7L. |
| 8 | Highest signal intensity at centromeric region, high at pericentromeric region, and weak at interstitial region of 8L. |
| 9 | High signal intensity at centromeric region and medium at interstitial region of 9L. |
| 10 | Highest signal intensity at centromeric region, weak at interstitial region of 10S, and medium at interstitial and subtelomeric regions of 10L. |
| 11 | Highest signal intensity at centromeric region, medium at pericentromeric and interstitial regions, and weak at subtelomeric regions of 11L. |
| 12 | Highest signal intensity at centromeric region and medium at pericentromeric and interstitial regions of 12L. |

similar to that of the centromeric SatDNAs of Type III in cucumber [14], pAL1 in *Arabidopsis* [29], pTS5 in *Beta procumbens* [30](Gindullis et al. 2001), and CentO in *Oryza* [31]. In this study, we conducted physical mapping of *Cmcent* and *CmSat162* on mitotic and meiotic chromosomes. *CmSat162* was located precisely at primary constrictions and flanked by the *Cmcent* signals, which are known to be located in centromeric regions. Although *CmSat162* and *Cmcent* share similar centromeric locations, the sequences of these two repeats are completely different (Fig 3a and S2 Table). These results indicate that *CmSat162* is a new centromeric repetitive sequence, and melon centromeres are composed not only of *Cmcent* but also of *CmSat162*.

*CmSat189* produced chromosome-specific signals not only at centromeric regions that were flanked by *Cmcent* but also at pericentromeric, interstitial, and subtelomeric regions. Using this probe, we elucidated its position and distribution in melon chromosomes. *CmSat162* and *CmSat189*, both of them were exclusively hybridized on the primary constrictions of melon. In addition, *CmSat162*, *CmSat189*, and *Cmcent* repeats shared low sequence homology (Fig 3a–3c), and their organizations shared high homology with *Cucumis melo* genomic scaffold sequence (accession number: LN681816) (S3 Fig). Thus, our findings suggest that *CmSat162* and *CmSat189* are new repetitive sequences for melon centromeres.

Although the centromere plays an important role in chromosome segregation during the cell cycle, its function is conserved and regulated by epigenetic mechanisms [32]. The expression of centromeric SatDNAs has been reported in plant species such as maize [33], *Arabidopsis thaliana* [34], rice [35,36], and banana [37]. The transcription of centromeric repetitive sequences, particularly SatDNAs, has essential functions not only in heterochromatin formation but also in maintaining centromere structures [26]. Bouzinba-segard et al. [38] reported minor satellite transcripts of 120 bp in murine cells that localize to centromeres. Forced accumulation of 120 bp transcripts leads to defects in chromosome segregation and sister-chromatid cohesion, changes in hallmark centromeric epigenetic markers, and mislocalization of

centromere-associated proteins essential for centromere function. Moreover, Rošić et al. [39] reported that the 359-bp satellite III (SAT III) of *D. melanogaster* was localized in centromeric regions of all major chromosomes and produced a long noncoding RNA. Depletion of SAT III RNA led to mitotic defects not only of the sex chromosome but also of all autosomes. SAT III RNA binds to the kinetochore component CENP-C and is required for correct localization of the centromere-defining proteins CENP-A and CENP-C, as well as outer kinetochore proteins. Thus, it is suggested that centromeric RNAs maybe play important roles in centromere function. In this study, *CmSat162* and *CmSat189* were shown to be actively transcribed in melon. These SatDNAs are maybe involved in the maintenance of melon centromere stability similar to those reported in maize by Gent et al. [40], where the authors revealed that diverse DNA sequences and multiple types of genetic elements in and near maize centromeres support centromere functions and constrain centromere positions. However, further analysis of the contributions of *CmSat162* and *CmSat189* to centromere stability is required to elucidate the specific functions of these two SatDNAs. Currently, there are no functional data of centromeric DNA repeats in *Cucumis* species, particularly in melon, that can be identified by chromatin immunoprecipitation (ChIP) using anti-CENH3 antibody followed by sequencing of the immunoprecipitated DNA (ChIP-Seq). The identification of melon centromeric region is solely conducted by FISH. Therefore, the ChIP-Seq is necessary to be conducted in the future to reveal functional centromeric repeats in melon.

## Supporting information

**S1 Fig. The consensus sequence of CmSat162 and is sequence alignment analyzed by Tandem Repeat Finder.**
(DOCX)

**S2 Fig. The consensus sequences of Cmsat189 and its sequence alignment analyzed by Tandem Repeat Finder.**
(DOCX)

**S3 Fig. *CmSat162* (A) and *CmSat189* (B) repeat organization in *Cucumis melo* genomic scaffold sequence (accession number: LN681816).** For *Cmsat162*, sequence from nt1141769 to nt1144288 of LN681816 was used for dot plot analysis whereas sequence from nt1875724 to nt1877241 was used for *CmSat189*. The consensus sequences of both CmSat162 and CmSat189 are listed in S2 Table.
(DOCX)

**S1 Table. SatDNA repeats on melon DNA sequence "LN681816" analyzed by Tandem Repeat Finder ver. 4.09.**
(PDF)

**S2 Table. The consensus sequences of *CmSat162* and *CmSat189* acquired from LN681816 DNA sequence analyzed by Tandem Repeat Finder ver. 4.09.**
(DOCX)

**S1 Row Images.**
(JPG)

## Acknowledgments

We would like to thank Henda Harmantia Dewi for editing and correction of this manuscript and Kousuke Kuroda for laboratory technical assistance.

## Author Contributions

**Funding acquisition:** Takato Koba.

**Investigation:** Agus Budi Setiawan, Chee How Teo, Hidenori Sassa.

**Resources:** Kenji Kato.

**Supervision:** Takato Koba.

**Validation:** Chee How Teo, Shinji Kikuchi, Hidenori Sassa, Kenji Kato.

**Writing – original draft:** Agus Budi Setiawan.

**Writing – review & editing:** Takato Koba.

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
