## [Decision Letter · Decision Letter 0]

4 Oct 2019

PONE-D-19-26046

Centromeres of Cucumis melo L. comprises Cmcent and two novel repeats, CmSat162 and CmSat189

PLOS ONE

Dear Dr. Koba,

Thank you for submitting your manuscript to PLOS ONE. After careful consideration, we feel that it has merit but does not fully meet PLOS ONE’s publication criteria as it currently stands. Therefore, we invite you to submit a revised version of the manuscript that addresses several points raised during the review process.

We would appreciate receiving your revised manuscript by Nov 18 2019 11:59PM. To enhance the reproducibility of your results, we recommend that if applicable you deposit your laboratory protocols in protocols.io, where a protocol can be assigned its own identifier (DOI) such that it can be cited independently in the future. For instructions see: http://journals.plos.org/plosone/s/submission-guidelines#loc-laboratory-protocols

We look forward to receiving your revised manuscript.

Kind regards,

Yamini Dalal, Ph.D.

Academic Editor

PLOS ONE

**Journal Requirements:**

**Comments to the Author**

1. Is the manuscript technically sound, and do the data support the conclusions?

Reviewer #1: Yes

Reviewer #2: Yes

2. Has the statistical analysis been performed appropriately and rigorously? 

Reviewer #1: N/A

Reviewer #2: N/A

3. Have the authors made all data underlying the findings in their manuscript fully available?

Reviewer #1: Yes

Reviewer #2: Yes

4. Is the manuscript presented in an intelligible fashion and written in standard English?

Reviewer #1: Yes

Reviewer #2: Yes

5. Review Comments to the Author

Reviewer #1: The manuscript “Centromeres of Cucumis melo L. comprises Cmcent and two novel repeats, CmSat162 and CmSat189” by Setiawan and others describes the identification of two new tandem repeat sequences. These two sequences largely colocalize with the established centromeric sequence Cmcent at the primary constriction. Melon centromere evolution is particularly interesting as species of this genus show evolutionary centromere repositioning events. These events are common in animals, but not in plants. What the role of centromeric sequences are in centromere repositioning events are remains unknown. Expanding our knowledge of what types of sequences are associated with centromeres, and whether these sequences are limited to the centromere or also found along the chromosome arms is important for understanding the evolutionary dynamic behavior of these fast evolving sequences. This work provides novel insights in satellite DNA in the very interesting Cucumis genus. Several concerns remain and are listed below.

+ The authors used scaffolds to identify new tandem repeats sequences using TRF. Could the authors provide the TRF settings used? Also, could the authors provide a reasoning why they chose to use scaffolds to identify novel tandem repeats, rather than WGS data? WGS data allows you to assemble from sequences that not only present in the scaffolds. Melters et al 2013 Genome Biology used this very approach to identify hundreds of candidate centromere tandem repeats. This approach would also provide the authors with an estimated quantity of the sequence identified.

+ Were the CmSat162 and CmSat189 sequences found the genome of other Cucumis species? A BLASTn search would give a quick indication.

+ What were the sizes of the FISH probes? In our hands, monomer sized FISH probes allow for very specific hybridization, whereas longer probes might hybridize with less specificity.

+ At the end of the introduction, the authors introduce the histone variant CENH3. For clarity, please specify that nucleosomes containing CENH3 instead of H3 mark the functional centromere.

+ In the final paragraph of the introduction, it is unclear what question the authors set out to answer in this manuscript.

+ Figure 1: did the authors also check for CmSat162 signal on pachytene chromosomes from ‘Baladewa’ and ‘Ivory F1 hybrids’? Ideally, this is reported as well.

+ Figure 2: did the authors also check for CmSat189 signal on ‘Baladewa’ and ‘P90’ chromosomes? Especially the karyotype ideogram is very informative. Showing this for all three cultivars and for both novel tandem repeats would provide a great resource for the field.

+ Do CmSat162 and CmSat189 colocalize on (pachytene) chromosomes of the three tested cultivars?

+ It is very interesting that transcripts of CmSat162 and CmSat189 were found by sqPCR. Were these transcripts cell cycle specific? Can this be determined from leave tissue? In addition, the -RT control, ideally a control with and without RNase, and with and without DNase is done to guarantee that no DNA contamination persisted.

Reviewer #2: This paper describes two satellites from melon that the authors show are cytologically located at the centromeres, and other locations. They also show that these satellite sequences are transcribed.

The presentation of the satellite sequences is somewhat confusing. The authors identify the 162 bp and 189 bp satellites using Tandem Repeat finder on a genomic scaffold. According to Table S1, the 162 bp repeat family and the 189 bp repeat family show 76% and 90% identity, respectively. They describe cloning the repeats using primers that amplify dimers of the repeats, but the primers given in the methods only partially match the satellite monomer sequences given in Table S2, making it unclear what sequences were actually cloned, and whether the monomers in Table S2 are particular examples of monomers or are consensus sequences. In Figure 3, comparing dimers of the two satellites to each other and to the previously identified satellite Cmcent, there is limited symmetry between the two halves of the dimers, especially for the 162 bp satellite, suggesting considerable divergence in the sequences of the two monomers used in the display. The authors also fail to mention how long Cmcent is and whether a monomer or dimer is displayed. They do mention in the Introduction that Sat107 is 352 bp long and is centromeric, but they do not identify this with Cmcent. There needs to be a better explanation of how the primers, monomers, and dimers relate to each other. The authors should state in the text in addition to Table S1 the degree of conservation of the two satellite families, and whether the sequences in Table S2 are consensus sequences or not. It might be useful to have a supplementary figure showing the relationship of the primers, monomer sequence in Table S2, and the dimer sequences in Figure 3 to the sequences amplified from the genomic scaffold.

Do the authors know whether one or both strands of the satellites are transcribed? Were the primers used for sqPCR the same as those used for gene amplification?

It would be helpful at some point for the authors to mention that there are currently no functional data or CENH3 ChIP data to identify the centromere sequences in Cucumis, and the identification is solely by FISH.

Minor points

The title is ungrammatical (Centromeres…comprises).

Line 3: Not all centromeres are associated with repetitive sequences. Though this is common in plants and animals, it is not

true of many single-celled eukaryotes.

Line 16: I think the authors mean “portion”, not “proportion”.

In Figures 2d and 3d, it would be helpful to the reader to number the chromosomes 1-12 at the bottom of the figures.

6. PLOS authors have the option to publish the peer review history of their article (what does this mean?). If published, this will include your full peer review and any attached files.

Reviewer #1: Yes: Daniël P Melters

Reviewer #2: No

---

## [Author Response · Author response to Decision Letter 0]

10 Dec 2019

We appreciate the comments of the editor and reviewers to the previous manuscript.

---

## [Decision Letter · Decision Letter 1]

23 Dec 2019

Centromeres of Cucumis melo L. comprise Cmcent and two novel repeats, CmSat162 and CmSat189

PONE-D-19-26046R1

Dear Dr. Koba,

We are pleased to inform you that your manuscript has been judged scientifically suitable for publication and will be formally accepted for publication once it complies with all outstanding technical requirements.

With kind regards,

Yamini Dalal, Ph.D.

Academic Editor

PLOS ONE

Additional Editor Comments (optional):

Reviewers' comments:

Reviewer's Responses to Questions

**Comments to the Author**

1. If the authors have adequately addressed your comments raised in a previous round of review and you feel that this manuscript is now acceptable for publication, you may indicate that here to bypass the “Comments to the Author” section, enter your conflict of interest statement in the “Confidential to Editor” section, and submit your "Accept" recommendation.

Reviewer #1: All comments have been addressed

Reviewer #2: (No Response)

2. Is the manuscript technically sound, and do the data support the conclusions?

Reviewer #1: Yes

Reviewer #2: Yes

3. Has the statistical analysis been performed appropriately and rigorously? 

Reviewer #1: N/A

Reviewer #2: N/A

4. Have the authors made all data underlying the findings in their manuscript fully available?

Reviewer #1: Yes

Reviewer #2: Yes

5. Is the manuscript presented in an intelligible fashion and written in standard English?

Reviewer #1: Yes

Reviewer #2: No

6. Review Comments to the Author

Reviewer #1: This reviewer acknowledges that all raised concerns have been commented on by the authors and adequately addressed most of the concerns raised. At the same time, this reviewer had hoped that the authors would be a little more forthcoming and elaborate more on some of the concerns raised or incorporate insightful information in the main text where feasible.

+ It would be helpful to the reader if the authors mentioned in the methods section that the default settings were used. Ideally in the response, the authors could mention why they think that the default setting was the appropriate setting to use.

+ The finding that CmSat162 and CmSat189 was only found in Cucumis melo and not in other Cucumis species is noteworthy of mentioning in the main text, as this might point to a recently evolved tandem repeat, or that the other Cucumis genomes are too incomplete to draw this conclusion. Using WGS data to identify novel tandem repeat sequences might be one approach to more confidently make such a statement.

+ The authors mention that Ivory F1 hybrids chromosome were not tested for CmSat162 and Baladewa and P90 chromosomes for CmSat189. It would be insightful if the authors could elaborate why they didn’t do this and why they think this is not important to do.

+ On the question about co-localization of CmSat162 and CmSat189, could the authors elaborate why they didn’t do this experiment and why they think it would add to their story?

+ Although not important for the story, this reviewer think it would be interesting to learn what they authors think about possible cell cycle specificity (if there is any) of the transcription of CmSat162 and CmSat189.

Reviewer #2: The authors have responded to most of the comments, but there are a few places where they still need to clarify what they have done. Reviewer 1 asked for TRF settings, which the authors provided in their reply but not in the Methods. They just need to add "with default parameters" to the mention of TRF in the Methods. Likewise they should state explicity that the same primers were used for genomic cloning and for sqPCR, and that the FISH probes are the 311bp and 327bp genomic clones. For example, on p. 6 they could say "using the same gene-specific primers of CmSat162 and CmSat189 used in cloning" and on p. 7 they could say that "the cloned PCR products for CmSat162 ad CmSat189 were labeled with Dig-Nick".

The primers described do not match the consensus sequences for CmSat162 and CmSat189 shown in S2 Table and appear to have been designed from the scaffold LN681816 (1142313-1142623 and 1876253-1876496). This should be mentioned where the primers are described. These primers give products of 311bp and 327bp, which are derived from dimers of divergent monomers, not from identical consensus monomers, which would yield products of ~145bp and ~55bp. The positions of the 311bp and 327bp clones could be added to S1 fig and S2 Fig, added to S2 Table, and/or given as accession numbers in a public database.

The authors leave the impression in their reply that Figures 3a and 3b display identical tandem dimers of the 162 and 189 consensus sequences ("we make a dimer of the consensus sequence"), but this does not seem possible, or else the left and right halves of these figures would be identical, which they clearly are not. Are the dimers in these figures taken from LN681816? Please clarify, and if so provide the specific coordinates for the dimers. Also the arrows showing where the primers are located would predict the 145bp and 55bp PCR products from monomers, not the 311bp and 327bp products that they actually amplified. It would be helpful to indicate where the primers actually bind, not where they would be expected to bind if the monomers were identical.

These clarifications can be easily fixed without further outside review, but they should be fixed, so that others can build on what the authors have done.

7. PLOS authors have the option to publish the peer review history of their article (what does this mean?). If published, this will include your full peer review and any attached files.

Reviewer #1: Yes: Daniël P. Melters

Reviewer #2: Yes: Paul B. Talbert

---

## [Editor Report · Acceptance letter]

30 Dec 2019

PONE-D-19-26046R1 

Centromeres of *Cucumis melo* L. comprise *Cmcent* and two novel repeats, *CmSat162* and *CmSat189*

Dear Dr. Koba:

I am pleased to inform you that your manuscript has been deemed suitable for publication in PLOS ONE. Congratulations! Your manuscript is now with our production department. 

With kind regards,

on behalf of

Dr. Yamini Dalal 

Academic Editor

PLOS ONE